# The Effects of Bituminous Binder on the Performance of Gussasphalt Concrete for Bridge Deck Pavement

**DOI:** 10.3390/ma13020364

**Published:** 2020-01-13

**Authors:** Guilian Zou, Xiaoyan Xu, Jianxin Li, Huayang Yu, Changjun Wang, Jie Sun

**Affiliations:** 1School of Civil Engineering and Transportation, South China University of Technology, Guangzhou 510640, China; huayangyu@scut.edu.cn; 2Petro China Fuel Oil Company Limited Research Institute, Beijing 100089, China; lijianxin@petrochina.com.cn; 3Petro China Fuel Oil Company Limited, Beijing 100089, China; wangchangjun@petrochina.com.cn; 4Highway Bureau of Transport of Shandong Province, Jinan 250000, China; sunjie602@163.com

**Keywords:** bitumen, gussasphalt concrete, deck pavement, performance, GMA, aging, Hong Kong–Zhuhai–Macao Bridge

## Abstract

As bituminous binders greatly influence the engineering performance of gussasphalt concrete for bridge deck pavement, selecting appropriate bitumen is a critical procedure for mixture design. In this study, five different combinations of bituminous binder for potential use in the “Hong Kong–Zhuhai–Macao Bridge” project were prepared and tested. To meet the strict requirements of quality control, a new Guss-Mastic Asphalt (GMA) system was developed. Three key indices, Lueer fluidity, impact toughness, and dynamic stability, were used for GMA design and construction quality control. The test results show that the fluidity of the GMA mixtures was affected by the shear thinning phenomenon. After mixing for three hours, the Lueer value of all GMA mixtures met the requirements of pouring construction. Moreover, it was found that the influence of mixing time on GMA with styrene butadiene styrene (SBS) modified bitumen was different to other prepared mixtures. This difference is ascribed to the degradation of SBS during the blending process at an elevated temperature. Finally, the blended bitumen (30% Pen60/70 + 70% Trinidad lake asphalt (TLA)) was applied to the project of the Hong Kong–Zhuhai–Macao Bridge.

## 1. Introduction

Gussasphalt concrete is one of the most widely used paving materials in steel deck pavement engineering because of its advantages including good adhesion, compliance, water resistance, durability, and self-leveling [1,2,3,4,5,6]. The gussasphalt concrete used for deck pavement can be divided into two categories: Gussasphalt (GA) and Mastic Asphalt (MA) [7,8,9,10]. GA has been widely used for waterproofing material in building construction and for pavement engineering since 1917. It has also been successfully applied in steel bridge deck pavement construction, such as in the Oberkasseler Bridge, Zoo Bridge, Mulheim Bridge, Cologne−Mulheim Rhine Bridge, Suderel Bridge, etc. In the United Kingdom (UK), based on the material’s characteristics, gussasphalt concrete was termed Mastic Asphalt (MA). The Forth Road Bridge was built in 1964 [11], and its deck pavement comprised a 38 mm MA layer, which had a service life of more than 30 years. The 38 mm MA deck pavement was also successfully applied to the Severn Crossing and Humber Bridge [12].

With the development of bridge deck pavement structures (with double-layer GA or MA and GA + Stone Mastic Asphalt (SMA) combinations), MA + SMA combinations have gradually been applied as composite pavement structure layers. Japan imported GA technology from Germany in 1956 and then made a great adjustment to the material composition and technical standard of German gussasphalt concrete gradually producing a set of technologies in accordance with Japan’s climatic and traffic conditions. A double deck pavement structure of 30–40 mm GA and 30–40 mm hot mix asphalt (HMA) concrete was adopted in the Akashi Kaikyo Bridge, Shiratori Bridge, Shizan Raido Bridge, Tataro Bridge, and other bridges [13,14]. In 1997, Taiwan first introduced GA technology to pave the bridge deck of the New East Bridge, and then the Gaoping River Bridge and Taibei Dazhi Bridge also adopted this technology, using a double-layer pavement material with a thickness of 80 mm (GA + modified asphalt HMA) [15].

The Hong Kong Tsing Ma Bridge, which was based on British MA paving technology, was opened to traffic in 1997 and has only been repaired twice on a small area [16]. The Jiangyin Yangtze River Bridge adopted the same technology, and opened to traffic in 1999. Rutting deformation and other problems appeared the following year, less than four years after the original bridge deck pavement had been replaced [17]. The Shenzhen Bay Highway Bridge and Stonecutters Island Bridge adopted MA + SMA double deck pavement which has been in service for more than 10 years and remains in good condition. To meet the strict requirements of quality control, a new Guss-Mastic Asphalt (GMA) system was developed, which combines the advantages of MA and GA technology, and was developed and applied for the construction of the Hong Kong–Zhuhai–Macao Bridge [18].

Bituminous binders influence the performance of gussasphalt concrete [19,20,21]. Germany, Japan, Britain, and other countries developed the technical requirements for binder selection, based on their respective climatic conditions, traffic environments, and usage habits. Trinidad Lake asphalt (TLA), a naturally occurring bitumen, has been widely used in gussasphalt concrete because of its low temperature sensitivity and good aging resistance [22]. Blended bitumen is commonly used in the United Kingdom. Ordinary bitumen with a penetration grade of 60/80 (0.1 mm) is mixed with TLA, and the content of TLA is generally 50–60%. In Germany, the blended bitumen used in GA mixtures is made from straight bitumen and TLA; the content of the TLA is 15–35%, and the penetration of the straight bitumen is 20–50 (0.1 mm). In Japan, blended bitumen is usually made from straight-run bitumen with a penetration grade of 20/40 and TLA, and the content of TLA is usually 20–30%. Although the Hong Kong Tsing Ma Bridge adopted a British bridge deck pavement material design system, because Hong Kong’s climate is hotter than that of the United Kingdom, the Tsing Ma Bridge increased the TLA content (70% TLA + 30% ordinary bitumen with a penetration grade of 60/80). The Shenzhen Bay Highway Bridge, Stonecutters Island Bridge, and Hong Kong–Zhuhai–Macao Bridge also used the same blended bitumen. 

With the development of bitumen modification technology, SBS modified bitumen has been used in some projects at home and abroad; double modified bitumen is sometimes mixed with SBS, while TLA is used as a binder. 

Due to the hot and humid climate in south China, a special design is necessary for the GMA mixture for the bridge deck of the Hong Kong–Zhuhai–Macao Bridge. In this study, five different combinations of bituminous binder were prepared and tested. The typical properties of this binder, including Lueer fluidity, impact toughness, and dynamic stability, were evaluated. In addition, the effect of the blending duration of bitumen and its aggregates was studied. This study is expected to provide information on the selection of bituminous binder for bridge deck pavement in hot and humid conditions.

## 2. Experimental Design

### 2.1. Materials and Sample Preparation 

#### 2.1.1. Bituminous Binders

The warm and humid climate in south China requires the high temperature performance of bridge deck pavement on the Hong Kong–Zhuhai–Macau Bridge. In summer, continuous high temperatures can cause volatilization, oxidation, decomposition, and polymerization of bituminous binders. In humid climates, the adhesion between bitumen and its aggregate is weakened due to water infiltration [23,24]. To ensure satisfactory rutting resistance, five different combinations of bituminous binder were selected and tested: 75% Pen20/40 + 25%TLA, 40% Pen20/40 + 60% TLA, 30% Pen60/70 + 70% TLA, SBS modified bitumen, and Pen15/25 hard bitumen. The combination of TLA and raw bitumen was achieved by high shear mixing at 240 °C for 1 h. Table 1 shows the basic properties of raw bituminous binders and the five combinations used in this study. Table 2 shows the test results of bituminous binders used in the study.

#### 2.1.2. Mixture Design

To meet the requirements of quality control and super-large-scale bridge deck pavement construction, GMA technology, which combines the merits of GA and MA, was adopted for the Hong Kong–Zhuhai–Macao Bridge.

The GMA mixture uses the British MA design system, which divides design into two stages, the ME design and determination of the ratio of coarse aggregate to MA. ME refers to the bitumen mastics combining the fine aggregate and bituminous binder. The coarse aggregate content in MA mixtures is specified to be 45% ± 10% by weight of the total mixture, for heavily stressed areas, and the soluble bituminous binder content will be between 14% and 17% by weight of ME. Table 3 shows the gradation of the MA mixture determined by a previous accelerated loading test on a test section of the Hong Kong–Zhuhai–Macao Bridge.

#### 2.1.3. Sample Preparation

A Cooker mixer that simulates the real heating and blending conditions of the GMA mixture in a bitumen plant was used for sample preparation. Figure 1 shows the equipment for the cooker simulating. In this study, the coarse and fine aggregates were preheated at 240 °C for 5 h before blending to prepare the bituminous binder. During mixture production, the bituminous binder was first mixed with fine aggregates (three minutes of blending) and then mixed with the coarse aggregates. The GMA mixtures with 1 h, 2 h, and 3 h blending durations were obtained for performance tests.

### 2.2. Testing Program

In the GA system, the fluidity of mixtures is used to evaluate the workability of gussasphalt concrete. The high temperature performance is characterized by an indentation test and a dynamic stability test. The MA system uses a hardness value, Marshall stability, and flow value to evaluate the high temperature performance. At present, the evaluation methods and indices of gussasphalt concrete in China are still in their exploratory stage. In this study, the properties of GMA were evaluated from the perspectives of fluidity and fatigue performance.

#### 2.2.1. Lueer Fluidity Test

Lueer fluidity has been widely used to characterize the workability of gussasphalt mixtures and is thus a reliable indicator to evaluate the fluidity of a GMA mixture. The Lueer fluidity test system consists of a container, a support frame, and a dombra-shaped drop hammer. The drop hammer was made of brass with a load of 995 g, and an indicator of 50 mm was marked on the upper part of the bar of the drop hammer. The time required to pass through the 50 mm indicator is called the Lueer fluidity at a specific temperature. The Lueer fluidity value of the GMA mixture at the 240 °C should be in the range of 4–20 s, according to the specifications for gussasphalt mixtures in China and Japan. The Lueer fluidity test follows the procedure in the technical guide for the design and construction of bridge deck pavement for highway steel box girders in China. An excessively low fluidity results in poor auto-compacting performance, and an excessively high fluidity makes the thickness and slope of the pavement uncontrollable. Three parallel tests were conducted in one group of tests. 

#### 2.2.2. Hardness Number Test

The MA system uses a hardness number test in accordance with BS 5284 to evaluate the ability of ME to resist deformation. The hardness number test mold is less than 100 mm in diameter and no less than 25 mm in depth. A flat-ended indenter pin in the form of a steel rod 6.35 mm in diameter is allowed to indent the mastic under a force of 311 N applied for 60 s while the mastic is maintained at 25 °C, and the depth of the indentation at the end of this period is recorded to an accuracy of 0.1 mm. Taking into account the differences in the climate environment and traffic volume between Hong Kong, China, and the United Kingdom, the test method and index of MA hardness values were modified when introducing MA technology in Hong Kong. The main processes of this modification are as follows. First, the test temperature is adjusted to 35 °C. Second, the MA mixture with a coarse aggregate is used in the test instead of ME, with the 1 min steady pressure deformation value of the MA mixture at 35 °C used as the hardness value. Third, the hardness number of MA as laid is 10–20 (0.1 mm) for the bridge deck pavement in Hong Kong. The hardness of the GMA mixture used in the Hong Kong–Zhuhai–Macao Bridge should meet the requirement of 5–20 (0.1 mm) to ensure satisfied rutting resistance under high service temperatures. Three parallel tests were conducted for one group of tests, and the indentation increment should not exceed 0.1 mm.

#### 2.2.3. Indentation Test

The indentation test was performed to evaluate the deformation resistance, following the procedure in the Chinese Test Specifications JTG D50-2017. For the indentation test, a cube specimen of 70 mm × 70 mm × 70 mm was maintained at 60 °C in a water bath for at least one hour. A load of 2.5 kg was initially applied to the specimen, and the zero point of the dial gauge was recorded. Then, a 52.5 kg load was applied smoothly. After 30 min and 60 min, the depth of the indenter was recorded to the nearest 0.01 mm. At least two repeated tests were performed, and the allowable deviation of the test results was less than 0.1 mm. The depth of the indenter recorded at 30 min was defined as the indentation of the test mixture, and the depth increment of the indenter between 30 min and 60 min was defined as the indentation increment of mixture. It is required that the indentation be in the range of 1–4 mm and the indentation increment should be less than 0.4 mm to ensure deformation resistance at high temperatures. Four parallel tests were conducted for one group of tests.

#### 2.2.4. Wheel Tracking Test 

The resistance to permanent deformation of the gussasphalt mixture was evaluated by a wheel-tracking test in this study, following the procedure in the Chinese Test Specifications JTG E20-2011. The GMA mixture slab with a dimension of 300 mm × 300 mm × 50 mm was prepared first. The testing temperature was 60 °C and the wheel load was 0.63 MPa. The dynamic stability (DS) was used to evaluate the rutting resistance of the GMA mixture. The minimum requirement of the dynamic stability is 300 cycles/mm for the GMA mixture. Three parallel tests were carried out, and the coefficient of variation between the three tests’ data should not be greater than 20%.

#### 2.2.5. Impact Loading Test

The impact loading test (ILT) has been developed and used in the Hong Kong–Zhuhai–Macao Bridge pavement to evaluate the fatigue characteristics of the GMA mixture The impact toughness parameter is used to characterize the fatigue capacity of the bitumen mixtures. The impact toughness parameter is derived based on the energy concept. As proven by a four-point bending beam test, there is a linear relationship between impact toughness and fatigue life, and the correlation degree is very high [25]. The ILT has low requirements for testing equipment and a short test period, so it is suitable for the fatigue performance index for design and construction control of bridge deck pavement. The test procedure involves first preparing a 300 mm × 300 mm × 50 mm plate specimen by the wheel milling method; secondly, 3–6 specimens are planned to be cut on the surface of the plate specimen formed by wheel milling, but a 20 mm portion of the edge is not used. A Finnish high-precision double-sided saw was used to cut the plate-shaped specimen formed by wheel milling into a prism beam with a span of 200 ± 0.5 mm, a length of 250 ± 2 mm, a width of 30 ± 2 mm, and a height of 35 ± 2 mm. The bitumen mixture prism specimen was placed in the water bath for more than three hours until the internal temperature of the specimen reached 15 ± 0.5 °C, and the beams were subjected to impact loading at a speed of 50 mm/min. The impact toughness value equals the area under a loading–displacement curve. The higher the impact toughness values of the materials, the better their ability to resist fatigue cracking.

## 3. Test Results

### 3.1. Lueer Fluidity Test

Table 4 shows the Lueer values of the test mixtures prepared with different blending durations. A lower Lueer value corresponds to better workability. It is noted that, for most mixture samples (except for the SBS modified bitumen mixture), the Lueer values decreased and then raised rapidly. This early decrement is ascribed to the thixotropy of the bitumen mixture. The shear thinning of the mixture gradually occurred with continuous blending, leading to superior fluidity. However, after blending at a high temperature for one hour, the influence of aging was dominant, the viscosity of the bitumen increased, and the fluidity of the mixture decreased gradually.

Table 4 shows the Lueer value variation of the GMA mixture for different blending times. As mentioned, the fluidity of the other bituminous mixtures was affected by the shear thinning phenomenon, and U-shaped curves can be observed. Based on Table 4, the GMA mixtures should have acceptable workability (with lower Lueer values over 20 s) with 2.5 h of blending time. Along blending time results in bitumen aging, which is detrimental to the workability of the GMA mixture, it is noted that the effect of aging varies for GMA mixtures with different bituminous binders. With a long blending time, the Lueer value increase of GMA with 40% Pen20/40 + 60% TLA was the most significant, followed by the 30% Pen60/70 + 70% TLA bitumen mixture. By comparison, the variation of the fluidity values for the SBS modified bitumen mixture and the Pen15/25 hard bitumen mixture were both under 20 s. Moreover, the GMA with SBS modified bitumen and Pen15/25 bitumen exhibited superior workability.

### 3.2. Hardness Number Test

Figure 2 shows the hardness of the GMA mixtures with different bituminous binders. We observed that a longer blending time resulted in a smaller hardness number. Moreover, ultra-high temperature blending ages the bituminous binder, resulting in a higher hardness for the GMA mixtures. When the blending time increases from one to two hours, the deformation resistance of the GMA increases sharply.

After mixing for more than two hours, the growth rate of the deformation resistance becomes slower. The change rule for the hardness value of the SBS modified bitumen GMA mixture is completely different. By increasing the blending time, the hardness number of the modified bitumen mixture increased first and then decreased. One possible explanation for this phenomenon is that ultra-high temperature aging causes the degradation of the SBS modified bitumen, which deteriorates the elastic recovery and capability of deformation resistance. With a higher aging level, the hardness value of each mixture becomes smaller.

### 3.3. Indentation Test

The GA from Germany used an indentation test (IT) to evaluate its ability to resist permanent deformation. The evaluation index included indentation and indentation increment, and the test temperature was 40 °C [26]. After GA was introduced into China, considering the high temperature in summer and the heavy traffic in China, the test temperature changed to 60 °C. The results of the indentation test are shown in Figure 3. In general, the IT results are in good agreement with those of the hardness number test. For most test samples, a longer blending duration leads to a higher aging level of the binder, which results in a lower indentation value. The only exception is the GMA with SBS modified bitumen. By increasing the blending duration, its indentation increased first and then decreased.

### 3.4. Wheel Tracking Test

The wheel tracking test (WTT) results of the GMA with different bituminous binders at different mixing times are shown in Table 5. With different mixing times, the DS of the GMA mixtures with different binders presented varying trends. The longer the mixing time under an ultra-high temperature (240 °C), the greater the DS of the GMA mixture with hard bitumen and blended bitumen. The dynamic stability trend of the modified bitumen GMA is completely different but remains consistent with the change trend for hardness and indentation. The SBS modified bitumen mixture’s ability to resist deformation decreased after mixing for one hour and increased again after mixing for more than 2 h. Ultra-high temperature mixing caused degradation of the SBS modifier and affected the aging process of the modified bitumen.

Hardness, indentation, and DS are the technical indices used to reflect the resistance of the mixture to permanent deformation. The effects of GMA for different bituminous binders based on these indices are basically the same. By comparison, the deformation resistance of GMA mixture with 40% Pen20/40 + 60% TLA was the strongest. Secondly, the deformation resistance of the GMA with 30% Pen60/70 + 70% TLA is close to that of the GMA with Pen15/25 hard bitumen, while the deformation resistance of the GMA with 75% Pen20/40 + 25% TLA was the worst. 

### 3.5. Impact Loading Test

The tensile strains measured at the top of the wearing course of the steel bridge’s deck pavement can exceed 500 µ strains, which is higher than the strains observed at the bottom of conventional bitumen pavement. This explains why most distress reported in the literature for orthotropic steel bridge pavement is related to the fatigue cracking of bituminous mixtures [27,28,29,30,31,32]. The impact toughness of the GMA mixture with different bituminous binders at different mixing times is shown in Figure 4.

According to Figure 4, the impact toughness of the GMA mixtures with hard bitumen or blended bitumen decreased by prolonging the mixing time, which indicates that the mixture gradually ages hardens, and becomes brittle; thus, the fatigue performance of the mixture decreases gradually. The impact toughness of the GMA mixture with SBS modified bitumen binder was higher than that of other GMA mixtures. Although SBS modifiers precipitate degradation at ultra-high temperatures, the residual SBS modified bitumen provides an enhancing certain effect on fatigue performance. According to its fatigue performance, the GMA mixture using 75% Pen20/40 + 25% TLA is better than that of the GMA mixture using Pen15/25 hard bitumen, followed by 30% Pen60/70 + 70% TLA and 40% Pen20/40 + 60% TLA.

## 4. Performance Balance

Thermo-oxidative aging results in structural damage and performance deterioration of the SBS modified bitumen [33,34]. It is noted that the blending duration has different influences on the performance of GMA mixtures with SBS modified bitumen compared to that of other GMAs. This is possibly ascribed to the aging of the SBS modifier during high-temperature blending. 

However, the mixing and paving temperatures of the gussasphalt concrete are generally 220 °C –230 °C, which leads to the rapid aging of the SBS modified bitumen and is, therefore, also referred to as super-heat aging. Because of these different aging mechanisms, the changing trends of the high-temperature performance indices (hardness, indentation and DS) of GMA with SBS modified bitumen are different from those of the GMA with other bitumen. Similarly, under the influence of SBS modified bitumen aging, the change trend of impact toughness reflecting the fatigue properties of GMA is also different from that of the GMA mixtures with other bitumen. Fourier transform infrared spectroscopy (FTIR) is used to recognize the related chemical bonds and functional groups.

The FTIR and XPS spectra indicate that the increments of aging temperature and aging time can accelerate the decomposition of tri-block structural SBS into bi-block structural SB-. Increments of temperature have a significant effect on the structural destruction of SBS in short-term aging [33]. Accordingly, polar hydroxyl and carbonyl groups appeared, and the C=C bond disappeared, in the molecular chain of the SBS [35]. The infrared spectrogram of the modified bitumen, as well as the super-heat aging of SBS for 0.5 h and 3 h SBS, is shown in Figure 5.

The aging of the SBS modified bitumen is divided into two parts: one is the degradation of SBS, and the other is the aging of the original bitumen. Super-heat aging over 200 °C causes the elastic recovery characteristics of SBS to degrade rapidly in the early mixing process. Therefore, the deformation resistance of GMA with SBS modified bitumen mixed for 1 h is only slightly better than that of ordinary hard bitumen. After 2 h of mixing, the aging of SBS modified bitumen is further deepened and deformation resistance continues to decrease. After 3 h of mixing, the aging effect of the original bitumen is dominant, and its deformation resistance ability begins to improve. In order to reduce the influence of thermal aging, the temperature of the hot mix modified bituminous mixture should not exceed 190 °C. 

## 5. Conclusions

According to the test results, blended bitumen (30% Pen60/70 + 70% TLA) should be used in the pavement project for the Hong Kong–Zhuhai–Macao Bridge. Blended bitumen (40% Pen30/40 + 60% TLA) and hard bitumen Pen15/25 are also feasible options. The price of hard bitumen is only about half that of TLA. Considering the price factor, hard bitumen Pen15/25 can be used as bitumen for bridge deck pavement in hotter areas. Typical properties, including Lueer fluidity, impact toughness, and dynamic stability, were evaluated. Comparing the ability of the anti-reflective cracking and the rutting performance test results, the change rule for the impact toughness is opposite to that of the indices, reflecting a general resistance to deformation. Therefore, in order to help balance both the deformation resistance and reflective cracking performance, we should choose a suitable rutting resistance index DS for the MA mixture because the rutting resistance index DS is not superior. An upper and lower limit is needed for the DS value. 

The DS of GMA should be controlled for 300–800 times/mm in the pavement engineering of the Hong Kong–Zhuhai–Macao Bridge. Three key indices are used to control the GMA’s quality. In addition to the requirement of DS, the impact toughness should not be lower than 400 N·mm, and the Lueer value should not be greater than 20 s. According to the test results, the following findings are summarized:Lueer fluidity is a reliable indicator to evaluate the fluidity of the GMA mixture. In the early stage of mixing, because of the shear thinning phenomenon, the Lueer fluidity-mixing time curve generally had a u-shape. The Lueer value of all bituminous mixtures can meet the requirements of pouring construction within 2–3 h.With an increase in mixing time, the deformation resistance of GMA is improved, and the results on the hardness, indentation, and dynamic stability follow the same trend. The deformation resistance of GMA with 40% Pen20/40 + 60% TLA was the strongest, followed by the GMA with 30% Pen 60/70 + 70% TLA, Pen15/25 hard bitumen and 75% Pen20/40 + 25% TLA. The indentation test can be used to evaluate the deformation resistance.The fatigue performance of the GMA mixture using 75% Pen20/40 + 25% TLA is the best among the test samples, followed by the mixture using Pen15/25 hard bitumen, 30% Pen60/70 + 70% TLA and 40% Pen20/40 + 60% TLA.The rutting resistance and fatigue performance were found to be inversely correlated. Based on the required performance balance, three key indices, Lueer fluidity, impact toughness, and dynamic stability, are suggested for GMA design and construction quality control.

## Figures and Tables

**Figure 1 materials-13-00364-f001:**
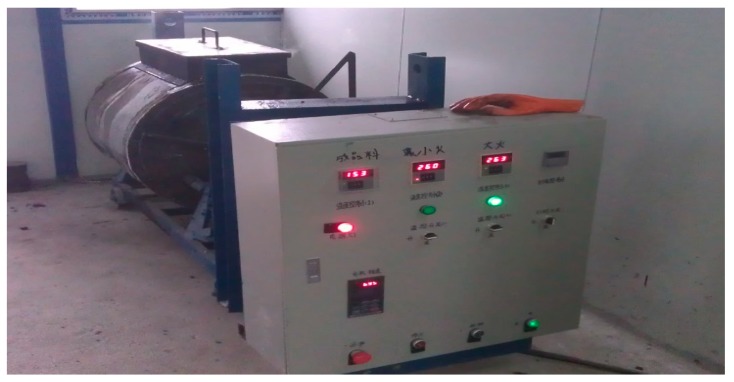
Cooker simulating mixer.

**Figure 2 materials-13-00364-f002:**
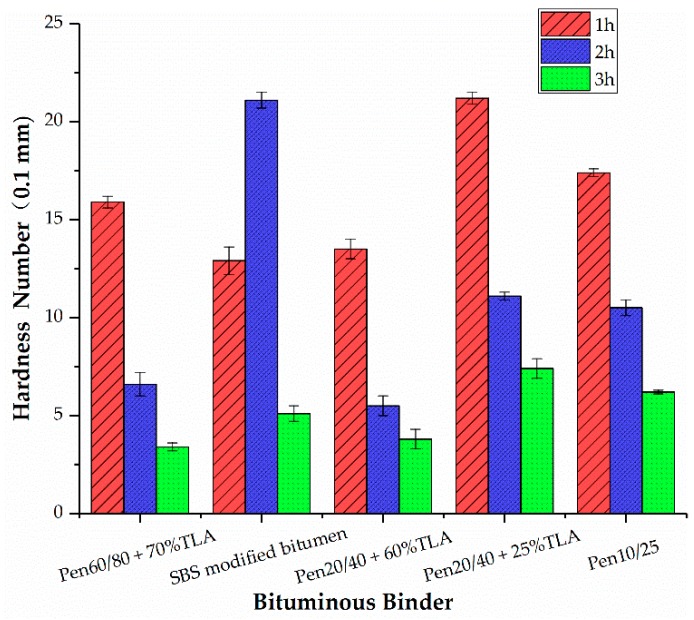
Effect of bituminous binder on hardness number for GMA.

**Figure 3 materials-13-00364-f003:**
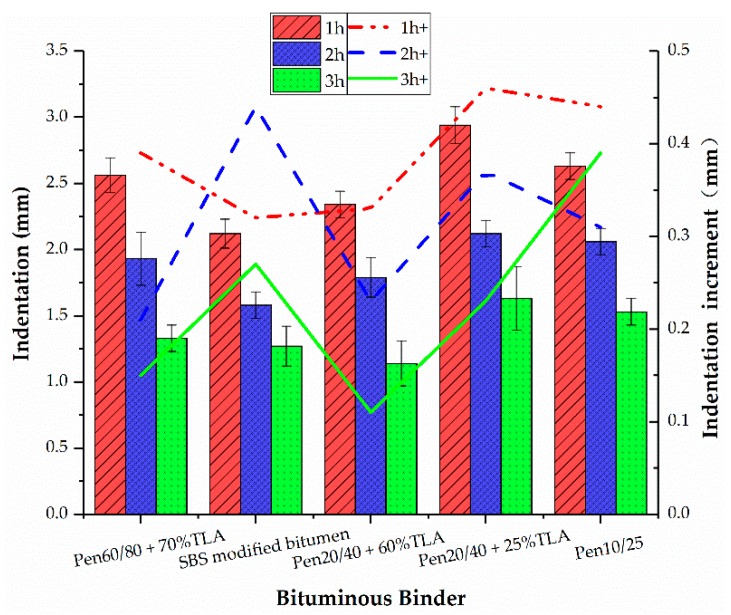
Effect of bituminous binder on the indentation test.

**Figure 4 materials-13-00364-f004:**
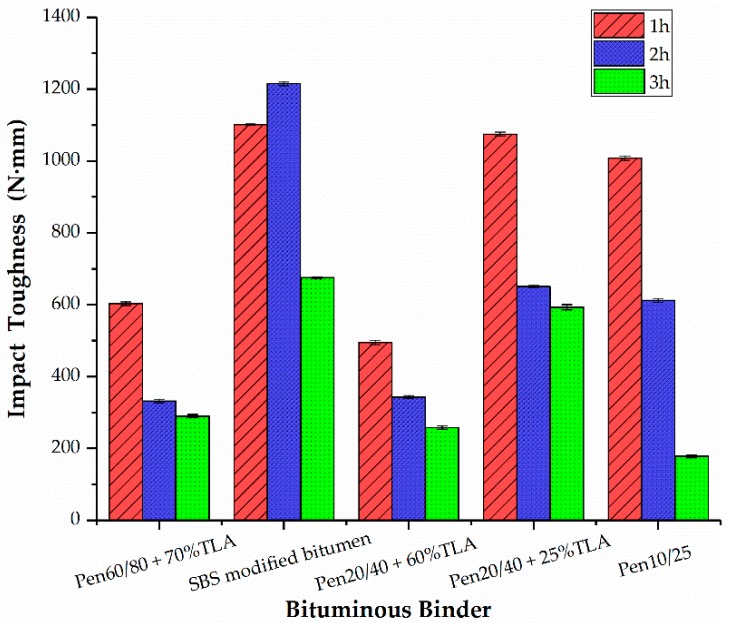
Effect of bituminous binder on the impact toughness values.

**Figure 5 materials-13-00364-f005:**
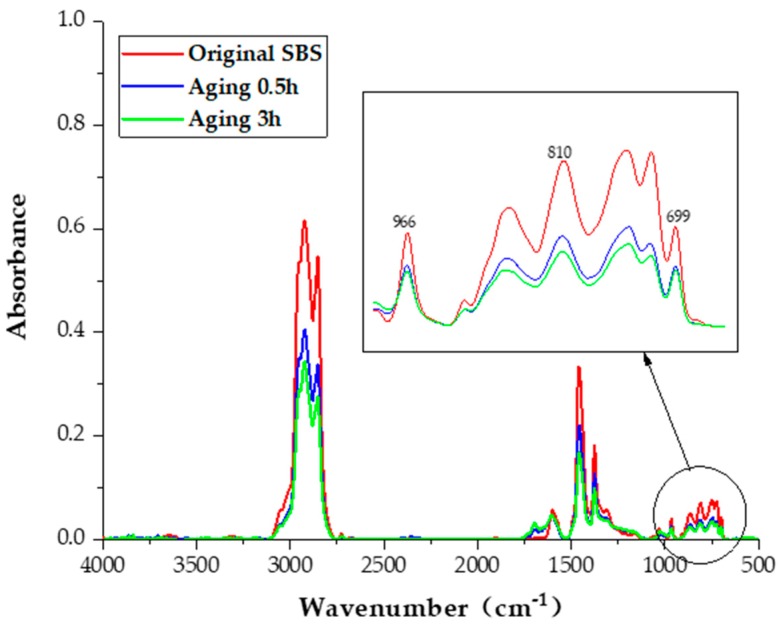
Super-heat aging of SBS in the FTIR test.

**Table 1 materials-13-00364-t001:** Results of bituminous binders before blending.

Indices	Unit	Pen15/25	Pen20/40	Pen60/70	TLA	SBS Modified Bitumen
Penetration at 25 °C	0.1 mm	18	36	63	3	53
Softening Point	°C	62	56	50.5	90	79
Ductility at 5 °C, 5 cm/min	cm	–	–	–	–	30
Ductility at 15 °C, 5 cm/min	cm	3	11	>100	–	–
Viscosity at 60 °C	Pa·s	–	1050	438	–	–
Viscosity at 135 °C	Pa·s	1.84	0.9	0.56	–	2.19
Total mineral matter	%	–	–	–	37.2	–
**Rolling Thin Film Oven Test (RTFOT) Residue (163 °C, 85 min)**						
Change of mass	%	−0.04	−0.04	−0.08	−0.95	−0.09
Penetration Ratio at 25 °C	0.1 mm	85.9	73.2	67.6	–	85
Ductility at 5 °C, 5 cm/min	cm	0.5	5	7.1	–	23
Ductility at 15 °C, 5 cm/min	cm	2	8	21	–	–

**Table 2 materials-13-00364-t002:** Test results of bituminous binders used in the study.

Indices	Unit	Pen15/25	75% Pen20/40 + 25% TLA	40% Pen20/40 + 60% TLA	30% Pen60/70 + 70% TLA	SBS Modified Bitumen
Penetration at 25 °C	0.1 mm	18	25	19	16	53
Softening Point	°C	62.0	62.5	70.0	73.0	79.0
Ductility at 25 °C	cm	68	36	14	13	>100
Viscosity at 135 °C	Pa·s	1.84	1.35	2.35	2.59	2.19
G*/sinδ at 64 °C	kPa	19.36	8.01	30.69	48.01	7.05

**Table 3 materials-13-00364-t003:** Gradation of MA mixture.

Gradation		% by Weight	BS1447:1988
**Percentage of Coarse Aggregate**		45	35–55
Passing sieve (%, sieves size / mm)	>2.36	0	0–2.5
	0.6–2.36	17	4–21
	0.212–0.6	25	8–32
	0.075–0.212	17	8–25
	<0.075	41	40–56
Soluble bituminous binder content in mastic epuré (ME)		14.5	14–17

**Table 4 materials-13-00364-t004:** Effect of bituminous binder type and mixing time on Lueer value.

Bituminous Binders	Lueer Values for the Following Mixing Time (s)			
	0.5 h	1 h	2 h	3 h
30% Pen60/70 + 70% TLA	9	6	11	28
SBS modified bitumen	5	8	10	14
40% Pen20/40 + 60% TLA	9	5	7	35
75% Pen20/40 + 25% TLA	8	7	9	26
Pen15/25	5	4	9	15

**Table 5 materials-13-00364-t005:** Effect of bituminous binder and mixing time on DS.

Bituminous Binders	DS at the Following Mixing Time (times/mm)		
	1 h	2 h	3 h
30% Pen60/70 + 70% TLA	378	662	>6000
SBS modified bitumen	491	316	1375
40% Pen20/40 + 60% TLA	384	2006	>6000
75% Pen20/40 + 25% TLA	233	616	2763
Pen15/25	372	863	2046

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
