# Peer review of "The Effects of Bituminous Binder on the Performance of Gussasphalt Concrete for Bridge Deck Pavement"

_materials, 2020, doi:10.3390/ma13020364_

Round 1

Reviewer 1 Report

General comments

This manuscript describes the laboratory testing related to a practical paving application. It is written in the form of a technical report. To become a scientific research paper, the Authors should add information in order to make their work more understandable and reproducible.

Specific comments

In the Abstract it is not necessary to specify the different combinations of binders (lines 17-18), TLA should be explained.

Line 44. SMA is not defined

Lines 43-61 Can you provide some literature references?

Line 65: “Trinidad Lake asphalt (TLA) has been widely used in gussasphalt concrete because of its low temperature sensitivity and good aging resistance.”. Reference needed.

Line 88: “The warm and humid climate in south China requires superior high temperature performance”. Please, just provide the technical specification to better explain the “superior” adjective.

Line 90: “75% Pen20/40 + 25%TLA, 75% Pen20/40 + 25% TLA” These two combinations are identical.

Line 92: “The combination of TLA and raw asphalt was achieved by high shear mixing at 240℃ for 1 hour.” How does this process affect short term aging?

Line 97: “To meet the strict requirements of quality control and super-large-scale bridge deck pavement construction” Please, just provide the “strict requirements”.

Line 111: “The equipment was designed and manufactured by Changda Highway Engineering CO. Ltd (Figure 1).” This is just advertising, please remove.

Can you provide some references on the Lueer Fluidity Test and the Indentation Test?

The “Impact Loading Test” is not well described. Method of compaction?

The description “the beam specimens were subjected to impact loading” is absolutely not suitable for a scientific research paper.

The Authors claim the existence of a relation between ILT and fatigue resistance. However, this relation is not proved. All the parts of the manuscripts referring to “fatigue resistance” should be removed.

Lines 177-180. This part is very general, thus is suitable for test description not for Results.

Table 4 and Figure 2 contain the same information, please remove one.

The “discussion” section contains the results of FTIR test. Why here? The method is not described in section

Author Response

Dear Reviewer:

On behalf of my co-authors, we thank you very much for giving us an opportunity to revise our manuscript, we appreciate you very much for their positive and constructive comments and suggestions on our manuscript entitled “ Effects of Asphalt Binder on the Performance of Gussasphalt concrete for Bridge Deck Pavement”. (ID: materials-673143).

We have studied the comments carefully and have made revision which marked in red in the paper. We have tried our best to revise our manuscript according to the comments. Attached please find the revised version, which we would like to submit for your kind consideration.

We would like to express our great appreciation to you for comments on our paper. Looking forward to hearing from you.

Thank you and best regards.

Yours sincerely,

xiaoyan xu

Reviewer 2 Report

1) In line 91, "SBS modified asphalt" is written, while in Table 1, 2, 4 and 5 as well as in Figure 2,3,4 and 5 in which properties of binders used in tests are compared, only the abbreviation "SBS" was given. The SBS symbol is a commonly used styrene-butadiene-styrene polymer denotation in the world. Therefore, using such a designation when comparing the properties of binders is misleading. Usually polymer-modified asphalt is designated as PmB. I suggest calling the binder as "bitumen" not "asphalt".

2) In the article there are some stylistic errors (e.g. lines 257-259) and spelling as well, e.g. line 103 should be "combining" not "combing" (nobody styled hairs) or instead of "indexes" should be indices (better refers to mathematical or scientific context).

3) Is it logically to name bitumen properties as  "technical indicators" ? Viscosity is a rheological parameter with physical meaning.

4) The authors did not provide any information regarding the number of samples and statistical analysis of test results. Therefore, the reliability of the test results cannot be evaluated. In Table 4 and Figures 3-5 authors repeatedly cite significant differences between the results. However, observing  slight differences between them, in some cases, conclusions presented by the authors are questionable. For example, on line 197 authors wrote "less noticeable". Did authors perform a ANOVA test to confirm this?

5) On bar charts, confidence intervals of results should be added (i guess bars represent a mean value). Comparing the results, the authors base their values on the average estimator, however without information on their variability, such evaluation the declared results is not sufficient.

6) Bar chart should be used in Figure 2. The line fit suggests that case denoted 40% Pen20 / 40 + 60% LTA has a minimum at 1.5 h (mixing time). Is it possible?

7) The authors presented the results of the binder study but did not specify the criteria that they should meet in order to be used for practical implementation.

8) The authors presented testing methods but did not refer them to AASHTO or EU standards, therefore it is difficult to assess whether the performed and obtained test results were universal or only local meaning.

9) What was the purpose of placing  Figure 6 at the end of the article without appropriate introduction. Absorbance spectra results cannot be used as significant without comparative analysis with other bitumen’s.

10) Conclusions are written too briefly and do not fully represent the content of the article.

Author Response

(The authors gave the same response as above.)

Round 2

Reviewer 1 Report

The added literature references in Response 3, 4 10 and 11 are not in suitable format (no source name, no publisher). I suspect the Authors did not check the references, which is not acceptable.

Section 2.2.1 is confusing, which one is correct:

“The Lueer fluidity value of GMA mixture at 240°C should be in the range of 4~20 s according to the specification of gussasphalt mixture in China and Japan”

“The required Lueer fluidity of GMA mixture is between 4 s and 40 s at 240°C.”

“To ensure the workability of the gussasphalt mixture, the Lueer fluidity value should less than 20 s”

Response 11 The Authors did not show any evidence that this “Impact loading test” is related to fatigue. In the response and in the manuscript (section 2.2.5) there is a lot of confusion among fatigue (resistance to cyclic loading), toughness (resistance to impact) and reflective cracking (resistance to crack propagation induced by a pre-existing crack). Fatigue was NOT investigated in this study.

Section 4 starts with a statement which looks like a conclusion: “if polymer-modified bitumen is used, it is suggested that warm-mix additive should be used together to reduce the degradation rate of SBS modified bitumen and help to control the performance of GMA”. Where does this statement come from? Why warm mix additives here?

In the rest of this section the Authors put:

citations from the literature (lines 317-330) description of the method (lines 330-332) experimental results (lines 333-339) discussion (lines 339-351)

Overall:

Overall, Section 4 looks like the summary of a different paper pasted here without special purpose.

The added parts (in red) contain several English language issues (e.g. line 353: “five different combinations of bituminous binder which potentially to be used in the project”)

Author Response

Dear Reviewer:   

We appreciate you very much for professional and constructive comments and suggestions on our manuscript entitled “ Effects of Asphalt Binder on the Performance of Gussasphalt concrete for Bridge Deck Pavement”. (ID: materials-673143).  

We have studied the comments carefully and have made revison. We have tried our best to revise our manuscript according to the comments. We would like to express our great appreciation to you for comments on our paper.

Looking forward to hearing from you.

Thank you and best regards.  

Yours sincerely,

xiaoyan xu

Reviewer 2 Report

The authors considered all reviewer comments and introduced them successfully in article.

Author Response

Dear Reviewer:  

On behalf of my co-authors, we appreciate you very much for professional and constructive comments on our manuscript entitled “ Effects of Asphalt Binder on the Performance of Gussasphalt concrete for Bridge Deck Pavement”. (ID: materials-673143).

We would like to express our great appreciation to you for comments on our paper. 

Thank you and best regards.  

Best wishes,

xiaoyan xu
